# HRM Accuracy and Limitations as a Species Typing Tool for *Leishmania* Parasites

**DOI:** 10.3390/ijms241914784

**Published:** 2023-09-30

**Authors:** Camila Patricio Braga Filgueira, Daniela Pitta-Pereira, Lilian Motta Cantanhêde, Gabriel Eduardo Melim Ferreira, Sayonara Dos Reis, Elisa Cupolillo, Otacilio C. Moreira, Constança Britto, Mariana Côrtes Boité

**Affiliations:** 1Leishmaniasis Research Laboratory, Oswaldo Cruz Institute, FIOCRUZ, Rio de Janeiro 21040-360, Brazil; camila.filgueira@fiocruz.br (C.P.B.F.); lilian.cantanhede@ioc.fiocruz.br (L.M.C.); elisa.cupolillo@ioc.fiocruz.br (E.C.); 2Laboratório de Biologia Molecular e Doenças Endêmicas, Instituto Oswaldo Cruz, FIOCRUZ, Pavilhão Leônidas Deane, Sala 209, Avenida Brasil 4365, Manguinhos, Rio de Janeiro 21045-900, Brazil; danypyta@gmail.com (D.P.-P.); cbritto@ioc.fiocruz.br (C.B.); 3Laboratório de EpiGenética, Fiocruz Rondônia, Rua da Beira, 7671, Porto Velho 76812-245, Brazil; gabriel.ferreira@fiocruz.br (G.E.M.F.); sayonara.reis@fiocruz.br (S.D.R.); 4Instituto Nacional de Ciência e Tecnologia de Epidemiologia da Amazônia Ocidental, INCT EpiAmO, Porto Velho 76812-100, Brazil; 5Laboratory of Molecular Virology and Parasitology, Oswaldo Cruz Institute, Fiocruz, Rio de Janeiro 21040-360, Brazil

**Keywords:** HRM, *Leishmania*, Americas, leishmaniasis, qPCR, parasite typing, HSP70, diagnosis

## Abstract

High Resolution Melting Analysis (HRM) has been pointed out as a suitable alternative method to detect and identify *Leishmania* species. Herein, we aimed to evaluate the sensitivity, specificity, accuracy, and limitations of a HSP70-HRM protocol both as a diagnostic scheme applied in clinical samples and as a species typing tool for laboratory research and reference services. Our data reveal the pronounced species-typing potential of the HSP70-HRM in DNA from cultured parasites. For clinical samples, however, we advise caution due to parasite load-dependent accuracy. In light of these findings and considering the importance of parasite load determination for clinical and research purposes, we recommend the integration of the presented typing scheme and the previously published *Leishmania* quantifying approach as combined tools for clinicians, surveillance, and research.

## 1. Introduction

Leishmaniasis is a worldwide, distributed, and neglected disease caused by protozoan parasites of the genus *Leishmania* [1,2,3]. The complex of diseases encompasses various clinical forms, from visceral to disfiguring cutaneous lesions [2,4]. Genetic diversity in the *Leishmania* genus plays an important role in the complex factors that lead to a wide range of clinical outcomes. In the Americas, the visceral form is caused by *Leishmania (Leishmania) infantum*, and tegumentary leishmaniasis (TL) is associated with the infection of 10 different species from the two subgenera *Leishmania* and *Viannia*. In Brazil, the most common species causing TL are *L.* (*V.*) *braziliensis, Leishmania* (*V.*) *guyanensis*, and *L.* (*L.*) *amazonensis*. In the Amazon region of the country, TL can also be caused by other species, such as *L.* (*V.*) *naiffi. L.* (*V.*) *lainsoni. L.* (*V.*) *shawi* and *L.* (*V.*) *lindenbergi* [5]. The species *L.* (*V*.) *utingensis*, first detected in sandflies [6] is also present in the region and has been recently detected infecting humans [7]. 

The diagnosis of leishmaniasis comprises the association of clinical, epidemiological, and laboratory data. Clinical signs and symptoms, alone or in combination, are not always sufficient as they can be confused with other diseases. It is therefore essential to carry out the laboratory diagnosis, which can be conducted by direct (parasitological, culture, histopathological analysis, and PCR) or indirect methods based on the detection of anti-Leishmania antibodies—less used for TL diagnosis [2]. The species diversity within the *Leishmania* genus has a major role in the treatment, clinical outcome, and transmission cycles. Therefore, the identification of the infecting species directly in clinical samples (preferably) or by isolating the parasite is still a relevant topic in the studies of leishmaniasis, especially in sympatric areas such as among countries and regions in the Americas. 

Biological collections gather valuable material to explore microorganism diversity and its impact on the development of diagnostic and species-typing tools. The Leishmania collection of Fiocruz (CLIOC; clioc.fiocruz.br) is a robust biobank, mainly for American *Leishmania* isolates, and is also a reference center for *Leishmania* species identification. The method applied for species typing is Multilocus Enzyme Electrophosis (MLEE)—still the gold standard technique but only applicable to successfully isolated and cultured parasites. MLEE has contributed hugely for typing [8] and for the epidemiology of leishmaniasis [9,10]; however, its drawbacks push researchers forward to the development of a cheaper and faster substitute method. Molecular-based approaches are currently the main target, and PCR-sequencing [11] and PCR-RFLP are already applied to DNA from cultured parasites and clinical material [12]. Both methods, however, involve a lot of steps, including gel electrophoresis. In this regard, protocols based on quantitative real-time PCR (qPCR) are good candidates because they eliminate the need for the detection of amplified products in gel, allow many samples to be assayed simultaneously, and have the potential to be automated. Moreover, it is especially relevant for clinical samples in which the parasite load affects sensitivity and accuracy, and the technique may potentially overcome this fact by detecting and quantifying minimal amounts of nucleic acids in a wide range of samples from different sources [13,14]. In this context, High Resolution Melting Analysis (HRM) has been pointed out as a good and practical alternative to detect and identify *Leishmania* species, both as a diagnostic tool and as a typing tool for laboratory work [15,16,17]. HRM is an adequate technique to detect mutations, single nucleotide polymorphisms (SNPs), and epigenetic differences in DNA samples. The analysis is based on the variance between the shapes of the melting curves and the difference between the melting temperatures (Tm). The Tm value of a sample can be obtained from the dissociation curve, defined as the temperature at which 50% of each DNA molecule is denatured [18]. 

A study developed by Zampieri et al. [19] presented a HRM assay using as targets two regions of the HSP70 *Leishmania* gene, both within a 234 bp region already used to identify species, even from clinical samples [12]. The first pair of primers targets a conserved region of 144 bp that amplifies all *Leishmania* species; a second pair targets a smaller and more variable region of 104 bp that identifies only *Leishmania* (*Viannia*) species [19]. Based on HRM analysis, these assays were able to differentiate eight *Leishmania* species found in the Americas: *L.* (*L.*) *infantum, L.* (*L.*) *amazonenses, L.* (*L.*) *mexicana, L.* (*V.*) *lainsoni, L.* (*V.*) *braziliensis, L. (V.) guyanensis, L. (V.) naiffi*, and *L. (V.) shawi*; and three species found in Europe, Asia, and Africa: *L.* (*L.*) *tropica. L.* (*L.*) *donovani* and *L.* (*L.*) *major*. The assays were validated in 16 experimentally infected golden hamster tissue samples, parasite strains isolated from humans, fresh tissue human samples or embedded in paraffin, experimentally infected BALB/c mice tissues, and samples from naturally infected phlebotomines. The HRM analysis results were compared to previous genotyping by sequencing and/or MLEE and presented a strong correlation between the methodologies. Finally, an algorithm was proposed to characterize by HRM the main *Leishmania* species circulating in Brazil [19].

Despite the potential offered by the published protocol [19], our group raised concerns on the following specific points: First, considering the well-known intraspecific genomic variation among *L.* (*Viannia*) strains, polymorphisms among distinct strains may affect the melting curve, melting temperature, and ultimately the HRM typing efficiency. Additionally, some species were not assayed previously, and neither was the intraspecific variation explored. Therefore, the effect of such diversity on the protocol must be investigated. By evaluating such aspects, we aim to appraise the potential of HRM to be applied as a substitute for MLEE. Secondly, an open question is how the accuracy, sensitivity, and specificity vary when the method is applied to a panel of human clinical material displaying a broad range of parasite loads. To address these subjects, we conducted a robust, distinctive, and representative sampling of *Leishmania* DNA available at CLIOC. Equally importantly, human clinical samples with associated previous data on the infecting *Leishmania* species and parasite load obtained by a HSP70 qPCR analytical study [20] were assayed by the protocol. The samples were collected from patients with cutaneous leishmaniasis from a region known to present high intra- and interspecific *Leishmania* diversity [20]. For the present study, we combine these described data and material, aiming to evaluate the accuracy of the HRM protocol [19], both as a diagnostic method and as a possible substitute for the MLEE as a species typing tool for the reference service. The approach allowed unique and comprehensive inferences to be made on the HRM potential as a species-typing tool and a diagnostic method for *Leishmania*. 

## 2. Results

### 2.1. HRM HSP70 Efficiently Identifies the Main Leishmania Species with High Reproducibility

Sixteen *Leishmania* reference strains representing 14 distinct species (Appendix A) were used for the adjustment of the HRM protocol. A subset of 13 samples was selected, and the reproducibility test was attained by two independent experiments performed by the same operator on different days. An absolute reproducibility (100%) of the variable profiles generated by the software was observed (Table 1). 

The typing outcome confirmed the separation of the main species circulating in Brazil, adding to the published information that *L.* (*V.*) *lindenbergi* and *L.* (*V.*) *utingensis* are also distinguishable by the algorithm. Clinically relevant co-circulating species in the Americas could not be distinguished using the strains evaluated in the present study: *L.* (*V.*) *braziliensis* vs. *L.* (*V.*) *peruviana*, *L.* (*V.*) *panamensis* vs. *L.* (*V.*) *guyanensis*. Based on the obtained data, representative samples were selected as references for further assays. The schematic representation of the variant profiles obtained and melting curves are summarized in Figure 1 and Figure 2, respectively. 

We further explored the possible effect of intraspecific diversity among *Leishmania* samples on HRM typing accuracy by assaying a larger and more representative panel of strains hosted by CLIOC. Appendix A depicts all strains used, including information on whether samples were applied as a reference, as validation, or during the reproducibility assays. All samples were previously characterized to determine the species by MLEE and/or DNA sequencing for the HSP70 target [11]. In total, 110 strains from distinct geographic areas of South America, mainly from the Amazon basin, were assayed, including three samples with hybrid profiles by MLEE and molecular typing methods (Table 2). To evaluate concordance between typing data, we excluded the references and the strains that could not be amplified by Primer 2. Results revealed concordance between MLEE/DNA sequencing and HRM typing data for 76/79 samples (96.2%). The HRM typing result of hybrid samples also converged with the other molecular typing approaches (Table 2). 

Kappa index indicated values compatible with “almost perfect agreement” (k = 0.91) between MLEE/DNA sequencing and HRM HSP70 typing results. All DNA samples presenting divergent outcomes (3/79) were sequenced a second time for the HSP70 region, and BLAST hits (ncbi.gov; accessed on 21 August 2023) confirmed the MLEE-based described species (Appendix A). These samples, thus, represent appropriate examples of the intrinsic limits of the HRM species typing protocol. 

### 2.2. HRM Discordant Results Cannot Be Directly and Exclusively Associated with Polymorphisms and/or Tms

The Tms distribution of both Primers 1 and 2 was color-coded and plotted according to the groups of variants (Figure 3), which were determined by the software based on the reference set of strains. The distribution of Tms portrays the fact that even samples with convergent results may present slightly distinct Tms from the references. This suggests that Tm is not the only parameter to define the melting curve profile and, ultimately, the species variants. The same conclusion applies for the occurrence of polymorphisms. Figure 4 and Figure 5 depict the polymorphic sites of representative samples, including the references and strains with concordant and discordant HRM typing outcomes. The Tms and polymorphism within the amplified region are detailed for each sample. 

Noteworthy, different combinations of various polymorphisms are observed among samples typed as the same variant, sharing similar Tm values, and melting curve profiles (e.g., Variant 1A). Conversely, the occurrence of very few polymorphisms was observed between samples with distinct Tms, melting curve profiles, and divergent typing results, as observed for the divergent IOCL 3310 (*L. naiffi*) strain for Primers 1 and 2 (Figure 4 and Figure 5). For this sample, only two SNPs (T/A and G/A) in the Primer 1 region led to Tm 85.7 °C, representing a 1.1 °C variation from the reference strain *L. naiffi* (IOCL 1365; Tm 86.8 °C). Distinct melting curve profiles were also observed for this sample (Figure 6). Primer 2 also revealed a divergent typing result for IOCL 3310 (*L. naiffi*) (Figure 5), despite the absence of polymorphism within the amplified region. The other two samples, both *L. braziliensis* strains, diverged in typing outcome; however, only for the second set of primers (Figure 5 and Appendix A). The polymorphisms observed in these samples (Figure 5) are the most likely cause for both the Tms and the melting curve profiles to diverge from the reference (Figure 6). These observations suggest that HRM divergent results cannot be directly and exclusively associated with polymorphisms and/or Tms, which, although combined, could potentially affect typing accuracy. Other undetectable factors may possibly be involved and would need further analysis.

### 2.3. HRM HSP70 Presents High Positivity and Similar Sensitivity and Specificity to the Previously Proposed HSP70 qPCR Protocol for Parasite Load Determination

To evaluate the accuracy of the HRM HSP70 protocol as a diagnostic tool, we assayed DNA obtained from clinical samples (n = 60) collected from patients presenting different clinical forms of tegumentar leishmaniasis. Samples had been previously submitted for microscopy, HSP70 qPCR-based parasite load determination [20], conventional PCR (cPCR), and typing by RFLP to determine the infecting *Leishmania* species (Appendix A). These previously published data allowed the current analytical assessment of HRM. 

The HRM positivity percentage obtained for these samples was 76.6% (46/60), the same obtained for microscopy (46/60), and slightly lower than that of the HSP70 qPCR (47/60; 78.3%). Conventional PCR (HSP70 cPCR) presented the highest positivity rate (81.6–49/60) for the same material. To determine the sensitivity and specificity, we plotted the data from the first set of HRM primers (P1) vs. microscopy and cPCR as a gold standard (Table 3). Further, we compared these parameters with those from HSP70 qPCR. The HRM assay presented higher sensitivity (82.6%) and specificity (42.9%) than HSP70 qPCR (80.4% and 28.6%, respectively) when microscopy was chosen as the gold standard. By employing cPCR as the gold standard, HRM sensitivity and specificity increased to 85.7% and 63.6%, respectively. These results suggest that HRM represents a good diagnostic tool for *Leishmania* detection, independent of species typing effectiveness. 

### 2.4. HRM HSP70 as a Species Typing Tool for Clinical Samples and the Effect of Parasite Load on Its Accuracy

We selected the cPCR-positive samples successfully typed by RFLP and/or DNA sequencing (45/60) to test the convergence with HRM typing results and thus evaluate its efficiency as a diagnostic-species tool. Among the 45 samples, 5 (5/45) revealed an inconclusive profile by the generation of singular melt curves distinctive from all reference strains; 6/45 were negatives, i.e., not amplified by Primers 1 or 2. Therefore, HRM species typing was achieved for 34/45 samples, among which 26/34 (76.5%) converged with the RFLP species typing result and 8/34 (23.5%) diverged (Table 4). Kappa index > 0 (K = 0.08) further confirmed the *Moderate Agreement* between RFLP data and HRM. This result on clinical samples is less accurate when compared to the successful outcome attained with DNA from cultured parasites (76.5% vs. 96.2%), an expected observation due to the nature of clinical samples of naturally infected individuals. 

Considering the potential effect parasite load has on molecular typing outcomes, we plotted the distribution of normalized parasite load in the samples according to the HRM-based species typing outcomes (Inconclusive, Divergent, and Converging). The result is represented in Figure 7. Most of the samples presenting inconclusive results by HRM (3/5) were likewise not quantifiable due to the low parasite load and therefore not represented in Figure 7. For the two remaining inconclusive samples, of which parasite load determination was possible, the median value of parasite equivalent per μg of human DNA was considerably higher (1.9 × 10^4^ par. Eq./μg human DNA) than for the Divergent and Convergent groups, respectively (2.8 × 10^1^ par. Eq./μg human DNA and 2.4 × 10^2^ par. Eq./μg human DNA). Such an association between typing success and *Leishmania* burden was also observed when Divergent and Convergent groups of samples were compared (Figure 7). We determine the quartiles and interquartile ranges to search for the parasite load interval that would provide better accuracy for HRM typing outcomes. The interquartile range (25–75%) of parasite load for all assayed samples was determined as 5.9 × 10^1^–6.0 × 10^2^ (par. Eq./μg human DNA) (Figure 7, doted lines). The Convergent group interquartile range was narrowed within 1.1 × 10^2^–5.7 × 10^2^(par. Eq./μg human DNA), while for the Divergent group. The range was widely dispersed outside this range (4.7 × 10^1^–3.3 × 10^3^ par. Eq./μg human DNA). Within the interquartile range (Figure 7, Quartile 2), 15/17 samples (88.2%) revealed an HRM convergent typing result, while for the first (Q1) and third (Q3) quartiles, the accuracy was reduced to 5/8 (62.5%) and 6/8 (75%), respectively. The data reveal the parasite load interval with the best accuracy for the HRM typing scheme.

## 3. Discussion

Our group has been interested in developing and validating methodologies to detect, quantify, and type *Leishmania* species, both as a laboratory/biobank protocol for the reference service and as a diagnostic tool to be applied in clinical material. The importance of detecting and typing *Leishmania* species is widely recognized, as is the determination of parasite load for clinical and research purposes [20,21]. Therefore, combined molecular approaches merging these possibilities would be valuable tools for clinicians, surveillance, and research. 

CLIOC, a *Leishmania* biobank, is currently evaluating techniques with the potential to replace the MLEE, aiming to overcome the drawbacks of the technique, such as its elevated cost, limited set of assayed samples per experiment, difficulty of inter-laboratory comparison, and time-consuming characteristics. High-resolution melting analysis is simple and rapid, and its use in clinical or research samples offers many advantages, such as a lower total cost for species identification compared to MLEE. Moreover, as a real-time PCR-based approach, there is no need for sequencing or gel electrophoresis to analyze the product, reducing the process and avoiding laboratory contamination with PCR products. It also reduces the need for trained personnel to analyze an electrophoretic gel or sequencing data to provide a result and, additionally, presents the possibility of quantifying infecting parasites within samples since it can be applied as a quantitative PCR-based technique. The whole process can be automated as the analyzer software produces the result by comparing the tested sample and the reference sample profiles, which must always be included in the reactions [22]. Based on this, we consulted the promising results from Zampieri et al. [19] and proposed to further evaluate the accuracy of the protocol as a typing and/or diagnostic tool.

First, we tested the reproducibility of the assay proposed by Zampieri and colleagues in our lab conditions, which include different operators and equipment, among other variables. The results of a reference set of samples fully reproduced the published data by differentiating the main species with the first set of primers (P1), except for *L.* (*L.*) *infantum* vs. *L.* (*L.*) *donovani* and *L.* (*L.*) *amazonensis* vs. *L.* (*L.*) *mexicana*. We included species not yet evaluated by the method, such as *L*. (*V*.) *peruviana*, *L*. (*V*.) *panamensis*, *L.* (*V.*) *lindenbergi*, *L.* (*V.*) *utingensis, L.* (*P.*) *hertigi*, *Paraleishmania colombiensis*, and *P. equatorensis*. The results precisely allowed the distinction of the main *L.* (*Viannia*) species, additionally revealing the profiles of the above-mentioned newly assayed species (Figure 1). By applying the second set of primers (variant 2), *L.* (*V.*) *guyanensis* and *L.* (*V.*) *braziliensis* were effectively differentiated, but *L.* (*V.*) *peruviana* vs. *L.* (*V.*) *braziliensis* and *L.* (*V.*) *panamensis* vs. *L.* (*V.*) *guyanensis* remained within the same variant and were, thus, not possible to distinguish. The separation of these species is especially relevant in sympatric regions such as the Andean region in Peru [23,24], Colombia [25,26], Venezuela [27] and Bolivia [28]. Further steps with an additional primer design must be developed to specifically address the purpose of typing these species. 

We performed a reproducible assay with the same operator, executing experiments on distinct days and obtaining 100% reproducibility. The next necessary step was to test the efficacy of the protocol, considering the intraspecific genetic variability known to occur among *Leishmania* strains. Such an assay allows for the evaluation of the effect of unanticipated DNA polymorphism on factors that affect the HRM typing outcome, such as mismatches at the primer alignment site, the melting curve, and ultimately the melt temperature [22]. Therefore, a wider panel of samples was prepared, composed of DNA strains from widely different geographic origins, representing 18 *Leishmania* species. Results converged with the MLEE and HSP70 sequencing data for 96.2% of samples, revealing a highly accurate outcome. The Kappa index indicated *Almost perfect agreement* between HRM and MLEE. Results revealed the potential for the proposed method and protocol to be applied as a typing tool on DNA from isolated and cultured *Leishmania*, eventually substituting the current MLEE assay. 

Raw data from the assays of the three divergent strains is a valuable opportunity to explore the intrinsic limiting features of the HRM approach. The melting temperature is one of the parameters used by the software to cluster samples within the window set defined by the reference strain profiles, though it is not the only metric. To express this subject, we plotted the Tm distribution with color-coded samples’ variants for both P1 and P2 primers. The result revealed Tm as not being the only parameter to define the melting curve profile, which ultimately defines the variants. Figure 3 summarizes the limited role Tm has on the clustering of samples as divergent and convergent. It has been reported that DNA methylation, concentration of the initial template for reaction—reflected by the Ct obtained, GC content within the amplified region, and stoic metrics—may influence the melt curve profile. Moreover, the type and quality of the DNA source material (purity and integrity), the isolation method, and pipetting inconsistencies might also influence the outcome [22]. We managed to carefully avoid a few of these variables to deliberately focus on the effect of polymorphisms on melt profiles. Thus, DNAs were prepared by the same isolation method within a six-month interval; all assays were performed by the same experienced operator on the same equipment. The concentration of the initial template for the reactions was not normalized, although parasites’ pellets for DNA isolation were prepared according to the same laboratory protocol. Therefore, the C_t_ values varied slightly, mostly within the described 20–30 range presented as optimal efficiency [22]. Indeed, the different quantities of the initial DNA template did not compromise the accuracy, whereas sequencing of the amplified region of the divergent strains revealed polymorphisms as an important, but not unique, source of divergence between typing data. These samples presented Tm close to the upper range of the correspondent references (Figure 3); however, the G-C content and the polymorphism position [22] were possibly the main factors that produced distinct melt curve profiles, ultimately leading to the divergent result (Figure 4 and Figure 5). The shape of melting curves may also be affected by methylation and efficiency in primer alignment, independently of polymorphisms [22]. However, for the current data, the polymorphisms and their location within the amplified region are the more plausible explanation for typing divergence. The intra- and interspecific genetic diversity is a well-known trait of *Leishmania* and indeed a challenge for species typing and for the development of diagnostic tools. There are geographic regions of Brazil where hosts and circulating parasites present greater chances of variability. Indeed, this was the case for the divergent samples from the validation step. These strains were isolated from dogs and from a human host in the northern region, which harbors a highly diverse parasite population [29,30]. 

Our group recently proposed an HSP70 qPCR-based approach to determine parasite load in material collected from patients with confirmed clinical diagnostics for cutaneous leishmaniasis. The patients studied were from an endemic area in the Amazon region, in which the parasite population is quite diverse [20]. In addition, the HSP70 qPCR, the samples were also subjected to microscopy and conventional PCR; infecting species were identified by PCR-RFLP and/or DNA sequencing. Herein, we applied the HSP70 HRM protocol to these samples to test the method’s accuracy as a diagnostic typing tool. Data revealed that the HRM positivity percentage was the same obtained for microscopy and slightly lower than that of the HSP70 qPCR. Conventional PCR (cPCR) presented the highest positivity. To define sensitivity and specificity, microscopy and cPCR were used as gold standards. The HRM sensitivity and specificity were higher than those obtained by the HSP70 qPCR applied to these samples. By employing cPCR as the gold standard, HRM sensitivity and specificity increased significantly. These results suggest HRM represents a good diagnostic tool for *Leishmania* detection, independent of species typing effectiveness. 

In contrast to the Kappa index compatible with to “*Almost perfect agreement*” and 96.2% concordance for the validation step in which DNA from cultured parasites was used, for the clinical samples, a “*Moderate Agreement*” was achieved, and convergent typing results were attained for 76.5% of samples. The effective outcome obtained by HRM with DNA from cultured parasites is expected due to the specific nature and purity of the samples. Even though predictable, we further explored the features underscoring the reduction in HRM typing success in clinical material. Coinfections, for instance, could lead to an inconclusive typing result, and such a possibility cannot be excluded given the geographic region where the patients were located (the sympatric area for most *Leishmania* species). These conditions, however, are difficult to detect and characterize. The most likely—and testable—factor involved is the parasite load. As described in [20], the clinical samples assayed presented a wide range of parasite equivalents per human DNA, thus comprising a suitable panel to be tested by HRM (Figure 7). Therefore, to evaluate the effect of parasite load on typing accuracy, we plotted parasite equivalents per human DNA according to HRM-based species typing outcome and defined the interquartile range of all samples. The result exposed the variable accuracy in each quartile, with the best outcome of 88.2% attained in a narrow range of parasite load. Therefore, HRM typing outcome was highly affected by parasite load, despite its overall high positivity for detecting *Leishmania* DNA. Based on this conclusion, we would suggest a clinical sample-typing workflow that first employs the HSP70 qPCR protocol [20] to evaluate whether the sample fits the most accurate parasite load range for HRM typing. The decision to further apply the HSP70-based HRM protocol as a typing tool must be taken considering this information. 

There are recent studies in the literature addressing the use of HRM to identify *Leishmania* species [17,31,32]. None, however, have tested the protocol accuracy in a significantly bona fide strains panel representing the inter- and intraspecific diversity of *Leishmania*. Important steps such as this are vital to proposing the technique as a typing tool. Moreover, the published studies avoid important validation steps by directly testing HRM-based protocols in biological material, not exploring the many factors that may influence the outcome. Herein, we reveal the great potential of the HSP70-HRM for species typing in DNA from cultured parasites. For clinical samples, however, the data raise the need for caution when using the HSP70-HRM protocol as a species typing tool. The variable accuracy dependent on parasite load imposes the need to first either determine the parasite load or, at least, establish a C_t_-based cutoff value before deciding on the typing approach. The typing accuracy was not tested in an artificially prepared DNA concentration range; instead, the variety and representativeness of the material from naturally infected individuals represented a considerably more realistic approach. 

The results presented herein open a venue of possibilities for evaluation of the HRM approach targeting other genes that might achieve better results related to *Leishmania* species identification, especially those targets with a higher copy number than the HSP70 gene. Among the different genes employed for *Leishmania* species typing, at least for neotropical regions, MPI [30] and Cytb [33] would be good alternatives to be tested. 

## 4. Materials and Methods

### 4.1. Strains and DNA Samples

All *Leishmania* strains were obtained from the *Leishmania* Collection of the Oswaldo Cruz Foundation (CLIOC—http://clioc.fiocruz.br, accessed on 1 August 2023). Parasite promastigotes were grown in a biphasic medium, constituted in the liquid phase by Schneider medium (Sigma, Chemical Co., St. Louis, MO, USA) supplemented with 20% inactivated Fetal Bovine Serum—FBS (Vitrocell, Campinas, SP, Brazil), and in the solid phase by 15% rabbit blood and BHI-Agar (Sigma, Chemical Co., St. Louis, MO, USA) [Novy–Nicolle–McNeal medium–NNN]. Strains were kept in the BOD incubator at 25 °C. The cell viability was evaluated by microscopy. Sixteen *Leishmania* reference strains representing 14 distinct species (Appendix A) were used for the standardization of HRM reactions. Additional strains representing the inter- and intraspecific genetic diversity of *Leishmania* circulating in the Americas were selected, including hybrids detected by MLEE (Appendix A). The following species not previously examined were assayed: *L.* (*V.*) *panamensis*, *L.* (*V.*) *utingensis*, *L.* (*V.*) *peruviana*, *L.* (*V.*) *lindenbergi*, *L.* (*Porcisia*) *hertigi, Paraleishmania colombiensis*, and *P. equatorensis.* In total, 110 strains representing 18 species were assayed. Additional data regarding samples, geographic origin, and hosts is available at the CLIOC website. The DNA was extracted from cultures using the Wizard Genomic DNA Purification Kit (Promega, Madison, WI, USA), according to the manufacturer’s recommendations. The DNAs were quantified in NanoDrop^®^ at 260 nm (Thermo Fisher Scientific, Waltham, MA, USA) and diluted in water to a concentration of 50 ng/µL. The DNA purity was estimated by the ratio 260/280 nm. 

### 4.2. Species Typing 

All samples from CLIOC are typed by Multilocus Enzyme Electrophoresis (MLEE) following the internal Standard Operational Procedures (SOPs). Additionally, for the present work, the HSP70 PCR products of the selected strains were obtained and sequenced, as previously reported [11], as an additional method to identify the species and reveal the polymorphisms within the region analyzed by HRM. The DNA sequencing of the HSP70 genomic region was performed by Fiocruz facilities. Briefly, PCR products were purified with the Wizard SV Clean-up System (Promega, Madison, WI, USA). The final DNA concentration was estimated by comparison with a DNA Ladder Marker (Promega, Madison, WI, USA) in a 1.5% agarose gel. Sequencing was performed with the same primers used for amplification using the BigDyeTerminator v3.1 Cycle Sequencing Kit, and the products were analyzed in an automated DNA sequencer (ABI PRISM-3730—Thermo Fisher Scientific, Waltham, MA, USA). Consensus sequences were generated from forward and reverse strands using the Phred-Phrap-Consed package [34]. Only sequences with Phred values above 20 were used for contig construction. 

### 4.3. HRM Reactions and Reproducibility Assay 

HRM reactions were designed to reproduce the protocol described by Zampieri et al. (2016), [19] due to the reported effectiveness of the method. A few technical variations, however, were included in the laboratory routine. Briefly, the equipment used was the ViiA 7 Real-Time PCR System (Thermo Fisher Scientific, Waltham, MA, USA) featured in a 384-well plate; MeltDoctor™ HRM Master Mix (Thermo Fisher Scientific, Waltham, MA, USA) was used at a 1x concentration per reaction in a 10 µL total reaction volume. Primer’s concentration was established at 200 nM after standardization steps. All experiments were performed twice, in technical duplicates. Dissociation analysis is conducted by capturing fluorescence signals at 0.2 °C intervals with the High-Resolution Melting Software (Thermo Fisher Scientific, Waltham, MA, USA). Protocols (primer 1 and primer 2, Table 5) were replicated for a larger sample panel of 110 strains, depicted above. The DNA of 16 reference strains (Appendix A) was assayed to first confirm the findings from the published study. To keep a technical or experimental replicate within the analysis, the silhouette score was considered. For each melt curve, a modified silhouette score is presented, ranging from 0 to 100. A score closer to 100 indicates that a melt curve is more like curves assigned to the same variant call than to curves called differently. For the reproducibility assay, 13 representative samples from the main panel were selected and submitted twice to the protocol in experimental duplicates by the same operator in two different days (depicted at Appendix A). 

### 4.4. Clinical Samples 

A panel of 60 DNAs from clinical samples was selected for the assays (Appendix A). The ethical recommendations of the Brazilian National Council of Health were adhered to, wherein a research protocol was duly registered and approved under the Certificate of Presentation for Ethics Appreciation code CAAE No. 0020.0.046.000–11 by the Ethical and Research Committee of the Center of Research in Tropical Medicine. The clinical material was collected by two sterile cervical brushes, used to scrape the TL patients’ lesions, and included positive and negative samples by microscopy and/or conventional PCR (cPCR) (Appendix A). Samples were stored in RNALater solution (Ambion^®^, Carlsbad, CA, USA) and kept at the Laboratory of Genetics Epidemiology from Fiocruz Rondônia (Fiocruz/RO), Brazil. The DNA was extracted using the PureLink Genomic DNA MiniKit (Invitrogen^®^, Carlsbad, CA, USA), according to the manufacturer’s instructions, excluding the incubation at 55 °C step. Identification of the species involved in the infections was performed by RFLP for the HSP70c target [11], using *Hae*III, *Bstu*I, and *MboI* enzymes [35]. As expected, most of them were identified as *L. (V.) braziliensis*, once the main species causing TL in that region. All clinical sample assays were performed blindly. The HRM protocol applied to these clinical DNAs was the same as previously described. 

### 4.5. Analysis 

Data on parasite load and species typing in clinical samples were obtained from our previous study [20]. Interquartile ranges of parasite load, Kappa index, sensitivity, and specificity were determined by GraphPad Prism v.9 (GraphPad Software, Inc., San Diego, CA, USA). Figures were designed in GraphPad Prism v.9 (GraphPad Software, Inc., San Diego, CA, USA) and Biorender (https://www.biorender.com, accessed on 8 August 2023). 

## Figures and Tables

**Figure 1 ijms-24-14784-f001:**
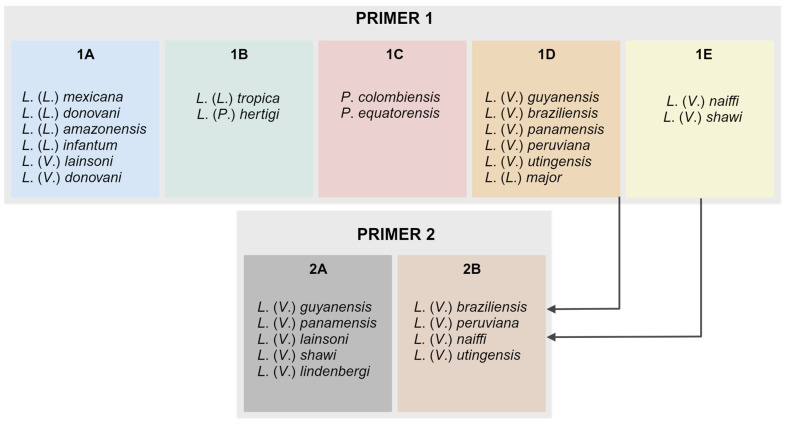
Schematic representation of the species groups coupled in each variant obtained by PCR reactions with primers 1 and 2; suggestion of the algorithm to differentiate the main species circulating in the Americas.

**Figure 2 ijms-24-14784-f002:**
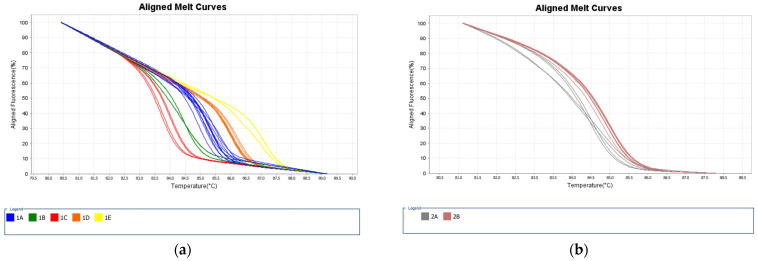
Color-coded dissociation curves of reference strains selected as standards for primers 1 (**a**) and 2 (**b**). (**a**) The identified variants for primer 1: blue—variant 1A; green—variant 1B; red—variant 1C; orange—variant 1D; yellow—variant 1E. (**b**) The identified variants for Primer 2: curves in gray—variant 2A; curves in brown—variant 2B. Figures were exported from the High-Resolution Melting Software (version 3.0) available at the ViiA 7 Real-Time PCR System (Thermo Fisher Scientific, Waltham, MA, USA).

**Figure 3 ijms-24-14784-f003:**
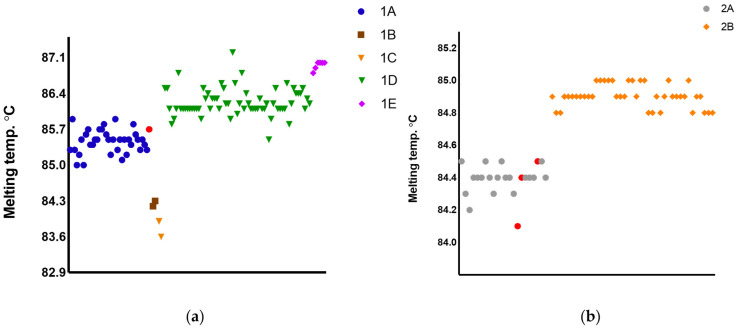
Color-coded Melting Temperatures (Tms) according to Primers 1 (**a**) and 2 (**b**) variants. TMs obtained during the validation assay (**a**): blue circle = 1A; brown square = 1B; orange inverted triangle = 1C; green = 1D; pink diamond = 1E. (**b**) Grey circle = 2A, orange diamond = 2B. The three divergent results are represented in red: (**a**) red circle: IOCL 3310, *L. naiffi* clustered as 1A (expected to be 1E); (**b**) red circles correspond to the two *L. braziliensis* and the one *L. naiffi* inaccurately clustered as variant 2A (expected 2B).

**Figure 4 ijms-24-14784-f004:**
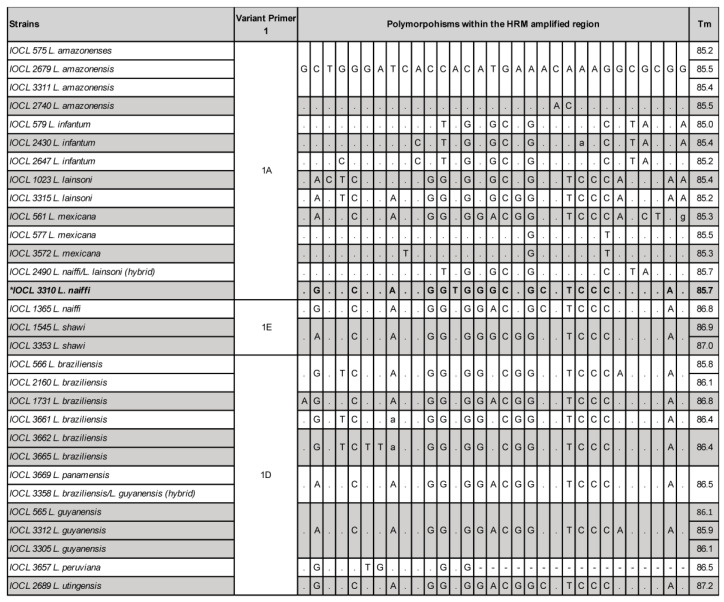
Examples of the variants obtained by Primer 1 for different strains and species include the polymorphisms observed within the HRM-analyzed region and the melting temperatures (Tm). The divergent strain IOCL 3310 is depicted in bold, marked *.

**Figure 5 ijms-24-14784-f005:**
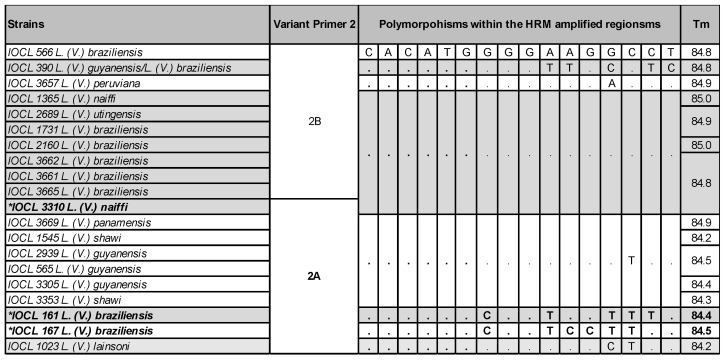
Examples of the variants obtained by Primer 2 for different strains and species include the polymorphisms observed within the HRM-analyzed region and the melting temperatures (Tm). The divergent strains IOCL 161, 167, and 3310 are depicted in bold, marked *.

**Figure 6 ijms-24-14784-f006:**
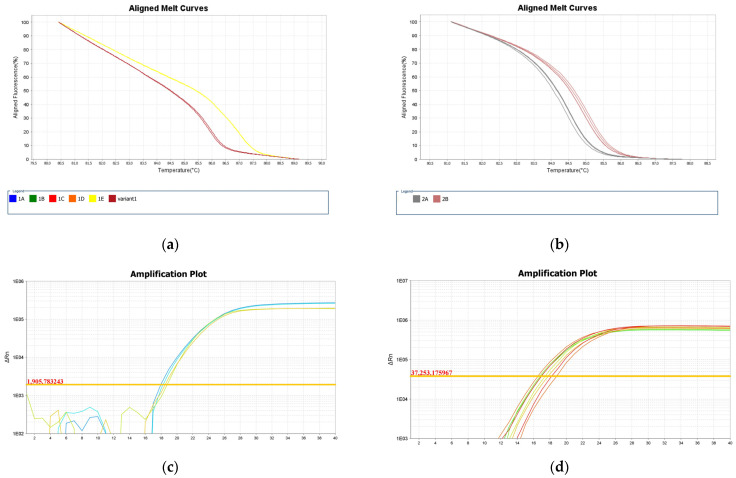
(**a**) Aligned melt curve profiles for the divergent strain IOCL 3310 (*L. naiffi*) (yellow), compared to the reference strain IOCL 1365 (*L. naiffi*) named by the software as a new variant (brown). (**b**) Aligned melt curve profiles and amplification plot for the *L. braziliensis* strains IOCL 161 and IOCL 167 obtained from Primer 2. The melt curves for both strains are represented in gray and were assigned as a different variant from the reference strain in red. (**c**,**d**) The amplification plot demonstrates similar amplification efficiencies for samples and the reference for primers 1 and 2. Figures were exported from the High-Resolution Melting Software available at the ViiA 7 Real-Time PCR System (Thermo Fisher Scientific, Waltham, MA, USA).

**Figure 7 ijms-24-14784-f007:**
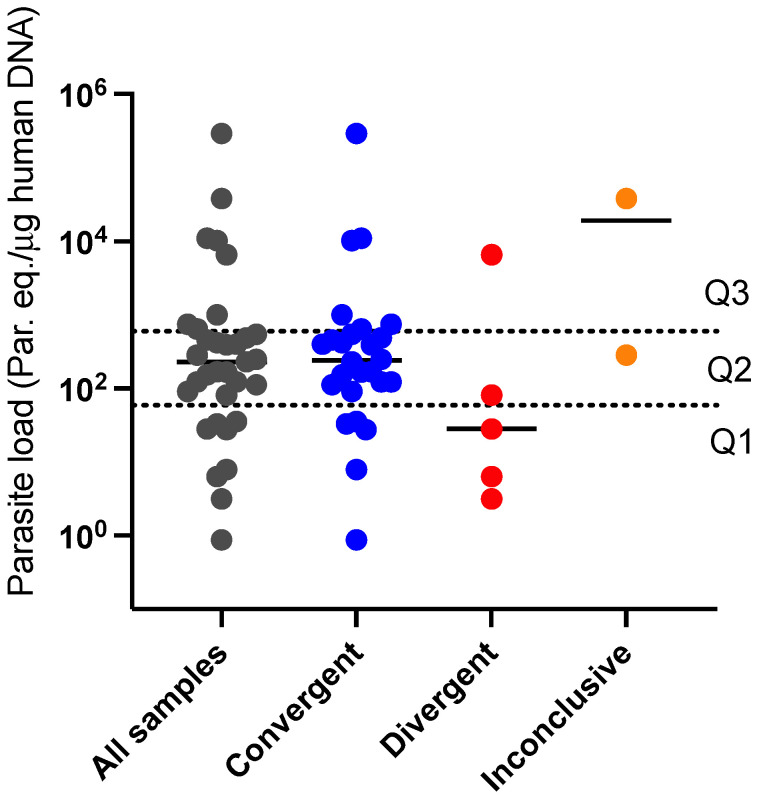
The normalized parasite load distribution of the samples assayed was plotted according to the species typing outcome. Convergent = HRM and RFLP convergent typing results. Divergent = HRM species typing was possible but did not match with previous RFLP characterization. Inconclusive = singular melting profile not correspondent to any of the reference strains. Doted lines mark the interquartile range of the total assayed samples, separating quartiles Q1 (bottom), Q2 (in between), and Q3 (upper area).

**Table 1 ijms-24-14784-t001:** Reference strains, typing results during experiments 1 and 2 for Primer 1 and Primer 2, the median of melting temperatures in °C (Tm) of replicates, and the standard deviation (SD) in °C.

Voucher	Species	Primer 1		Primer 2		Primer 1		Primer 2	
		Exp. 1	Exp. 2	Exp. 1	Exp. 2	Tm	SD	Tm	SD
IOCL 565	*L. (V.) guyanensis*	1D	1D	2A	2A	86.1	0.2	84.1	0.0
IOCL 566	*L. (V.) braziliensis*	1D	1D	2B	2B	85.8	0.1	84.7	0.1
IOCL 575	*L. (L.) amazonensis*	1A	1A	-	-	85.2	0.1	83.9	0.2
IOCL 579	*L. (L.) infantum*	1A	1A	-	-	85.0	0.1	85.1	0.0
IOCL 1023	*L. (V.) lainsoni*	1A	1A	2A	2A	85.4	0.1	84.2	0.1
IOCL 1365	*L. (V.) naiffi*	1E	1E	2B	2B	85.4	0.1	84.7	0.2
IOCL 1545	*L. (V.) shawi*	1E	1E	2A	2A	86.7	0.1	84.0	0.1
IOCL 2689	*L. (V.) utingensis*	1E	1E	2B	2B	87.0	0.1	84.8	0.2
IOCL 2690	*L. (V.) lindenbergi*	1D	1D	2A	2A	85.9	0.1	84.1	0.1
IOCL 3394	*L. (V.) braziliensis*	1D	1D	2B	2B	86.1	0.1	84.6	0.0
IOCL 3398	*L. (V.) lainsoni*	1D	1D	2A	2A	85.7	0.2	84.1	0.0
IOCL 3399	*L. (L.) amazonensis*	1A	1A	-	-	85.2	0.3	84.6	0.0
IOCL 3538	*L. (V.) guyanensis*	1D	1D	2A	2A	86.2	0.1	84.2	0.0

1D: variant 1D; 1A: variant 1A; 1E: variant 1E; 2A: variant 2A; and 2B: variant 2B (see Figure 1). Exp. 1 and Exp. 2 refer to the experimental replicates 1 and 2 performed by the same operator on different days.

**Table 2 ijms-24-14784-t002:** Samples with hybrid profiles by MLEE and respective variants and Tm obtained by HRM.

V 1	V 2	Voucher (IOCL)	MLEE Characterization	DNA Sequencing	HRM	Tm V1	Tm V2
1D	2A	3358	*L. braziliensis/L. guyanensis* (hybrid)	*L. guyanensis*	*L. guyanensis*	86.5	84.1
1A	2A	2490	*L. naiffi/L. lainsoni* (hybrid)	*L. lainsoni*	*L. lainsoni*	85.7	84.3
1D	2B	390	*L. guyanensis/L. braziliensis* (hybrid)	*L. braziliensis*	*L. braziliensis*	86.5	84.8

V-1 variants were observed after Primer 1 amplification. V-2 variants were observed after Primer 2 amplification.

**Table 3 ijms-24-14784-t003:** Sensitivity and specificity values for Leishmania detection by HRM (P1) and by HSP70 qPCR, compared to cPCR and microscopy as gold standards.

Assays	Patient Samples (n = 60)		
	HRM +	HRM −	Total
Microscopy +	38 (82.6%)	8 (17.4%)	46 (100%)
Microscopy –	8 (57.1%)	6 (42.9%)	14 (100%)
cPCR +	42 (85.7%)	7 (14.3%)	49 (100%)
cPCR –	4 (36.4%)	7 (63.6%)	11 (100%)
Assays	HSP70 qPCR +	HSP70 qPCR −	Total
Microscopy +	37 (80.4%)	9 (19.6%)	46 (100%)
Microscopy –	10 (71.4%)	4 (28.6%)	14 (100%)
cPCR +	40 (81.6%)	9 (18.4%)	49 (100%)
cPCR –	4 (36.4%)	7 (63.6%)	11 (100%)

**Table 4 ijms-24-14784-t004:** Results from HRM HSP70 as a species typing tool in clinical samples.

HRM Outcome	Samples Successfully Typed by RFLP (n = 45/60)
TOTAL	45 (100%)
Inconclusive	5/45 (11.2%)
Negative	6/45 (13.3%)
Species typing achieved	34/45 (75.5%)
HRM outcome	Samples with HRM species typing data (n = 34/45)
Total	34 (100%)
Divergent typing result	8/34 (23.5%)
Convergent typing result	26/34 (76.5%)

**Table 5 ijms-24-14784-t005:** Primers used for HRM reactions. Amplicon sequences and references.

ID	Sequence	Fragment	Reference
P1	HSP70 F2 5′-GGAGAACTACGCGTACTCGATGAAG-3′ HSP70C R 5′- TCCTTCGACGCCTCCTGGTTG-3′	144 pb	Zampieri et al. 2016 [19]/Graça et al. 2012 [12]
P2	HSP70 F1 5′-AGCGCATGGTGAACGATGCGTC-3′ HSP70 R1 5′-CTTCATCGAGTACGCGTAGTTCTCC-3′	104 pb	Zampieri et al. 2016 [19]

## Data Availability

Data supporting the reported results on the *Leishmania* samples available at CLIOC can be found at https://clioc.fiocruz.br (accessed on 8 August 2023).

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
