# Peer review of "HRM Accuracy and Limitations as a Species Typing Tool for Leishmania Parasites"

_ijms, 2023, doi:10.3390/ijms241914784_

Round 1

Reviewer 1 Report

This manuscript, which proposes an HRM protocol for typing Leishmania, includes numerous samples and provides a lot of data and assays. 

This study is quite similar to the one on which the authors based their work, with a few additional new elements.

Major remarks:

Bibliographical references :
references 1 and 7: it would be preferable to include the WHO reference
reference 3: perhaps also include the WHO reference
reference 19: this is the same reference as 18.
Line 442, verify that reference 9 is the correct one
Line 450, verify that reference 20 is the correct reference
Line 454, verify that reference 17 is the correct reference
Line 491, verify that reference 19 is correct
Line 485: verify that reference 34 is the correct reference, it appears to be 10

Line 179: in the legend, it should be replace black with grey and blue with orange. Perhaps also add, on the line 177: (a) after "Primer 1" and (b) after "Primer 2", to make the figure easier to read.

Line 196: If we compare the strain 3310 with the reference strain L. naiffi 1365, don't we also observe a G/A SNP with primer 1 as well? Has the rest of the amplified sequence been analysed for polymorphism? is there also polymorphism in these other parts?

Line 199: It is indicated that there is no polymorphism in the region studied. However, only one region is shown in Figures 4 and 5. Has the rest of the amplicon been observed and does it show polymorphism?

This study suggest a protocol for discriminating/typing species, and for this purpose groups are shown in Figure 1. Have "box and whisker plot" type graphs been produced to see how the Tms of the different species overlap, particularly those between variants? If so, are these overlaps significant and do they show the possibility of making it difficult to interpret results even between variants?

Minor comments:

Line 119: is it 9 different species analysed rather than 13?

Table 1: the legend indicates the different variants. It would be useful to indicate what these variants correspond to by making a reference to a figure or something else, to help understand what they are referring to.

Figures 2b and 6b: the small yellow box in the legend should be removed.

Line 144: does the analysis of Leishmania refer only to the L. braziliensis species? Perhaps specify if this is the case.

Lines 147-148: How was the choice made to decide whether a strain was used as a reference, validation or reproducibility?

Line 149: it is indicated that 110 strains were used for the study. Are these the ones presented in table S2? If so, there seem to be more.

Line 150: it is indicated that the strains come from different geographical regions. Did the differences in HRM profiles show a link with the geographical area from which the Leishmania strain originated? Also, is the polymorphism shown on lines 191-192 associated with particular geographical areas?

Line 154: it might be indicated why do we go from 110 samples to 79?

Line 181-182: can we really talk about an error?

Figure 6b: in the graph legend, what do the 3 variants shown correspond to?

Figure 6c and d: it would be necessary to specify a little more precisely what is to be shown with the effectiveness of PCR.

Line 240: it is stated that 60 DNA samples were studied. Does this number of samples correspond to those indicated in table S3? if that's the case, aren't there more?

Line 241: in the sentence "among the samples…"  it might be specify "among these samples" or "among the 45 typed samples"?

Lines 247 and 248: it might be indicated the number of samples (n/n) corresponding to the percentages indicated (76.5% and 96.2%).

Line 256: the full name "par.eq./µg" could be specified initially.

Line 266 : it should be verified that the 4.7 x 10-1 value is correct 

It is indicated that the parasite load seems to have an impact on HRM results. Has a range of DNA concentrations been used to determine the limit above which the results are unreliable? This also relates to the conclusion made on lines 411-412: has a limit been determined?

Table 4: the last line of the table shows a total of 45 (100%), does it correspond to the data at the top of the table ?if this is the case, the total should be move up.

Line 325: it is stated that 90 strains were studied. Is this correct? because throughout the text it is indicated 110 strains (line 140, 432 and 462).

Line 326: it is stated that 19 species were studied. Is this correct? because in the materials and methods (line 432), it is indicated 18 different species. Although, if we consider lines 430-432, it would appear that there are 21 different species.

Line 364-365: it is indicated that the strains were isolated from dogs and humans. Are these CLIOC strains? If so, it would be useful to indicate this in the material and method, part 4.1 (lines 420-421).

Line 465-470: this paragraph explains how species are distinguished on the basis of HRM results. Then, is this the description for obtaining the different variants shown in figure 1? If this is the case, it could be indicated.

Table 5: the references have been given in the form of names, to make it easier to find bibliographic references, the reference numbers could also be added: [18], [11].

Line 472: optionally put the part "4.4 clinical sample" after "4.1 strains and DNA samples"

Line 482: isn't the name of the complete extraction kit "PureLink Genomic DNA" kit? if so, perhaps complete the name.

Line 485: It is stated that RFLP was carried out on the HSP70c target. Was the HSP70c target amplified with the P1 primers? If so, it could be indicated.

Supplementary data :

Table S1: a legend could be added to explain V1 and V2 (and a reference to a figure or other to explain the different variants).

Table S2: In the legend, before writing "Voucher or strains...", perhaps introduce "the table includes the voucher or strains...". Does DESP corresponds to the standard deviation ? perhaps write DESP in the legend. It should be indicated the meaning of SA and NA. Optionally, indicate what a validation/reproducibility test is and why a strain is considered to be a reference or not ( related to the comment above). Correct HRM Variant 2 in the first line of the table.

Table S3: In the legend, perhaps add "for Primer 1 and Primer 2" after "by the current HRM approach".

A little more explanation is needed for "concomitantly assayed as reference and as DNA form cultured promastigotes". Why are some samples named IOCL and not others, are they all IOCL (or is it to indicate those that are "concomitantly assayed as reference and as DNA form cultured promastigotes")?

In a large part of the manuscript, the genera/species are not italicised. IOCL should be replaced by CLIOC to remain uniform (in the text, tables and figures) or everything should be put under IOCL.

Here, a few typos in punctuation (lines: 32, 36, 40, 134, 258, 336, 460, 487), or spelling (polymorphism lines 191 and 193, enzyme line 485), and other : line 439 (it can be put "were" instead of "are"), Figure 4 and 5 (correct "regions"), line 256 (rather put a lower-case "e" in front of "Eq.",as well as in the rest of the text, and leave par.eq./µg without brackets).

Reviewer 2 Report

The authors have described and validated an existing HRM method for detecting and identifying Leishmania species. The article is well-written, and the validation itself is solid. However, it would be beneficial and crucial if the authors include a comparison with the previous HRM method that did not include the modifications introduced by them in the "Materials and Methods" and "Results" sections. Additionally, a "Conclusion" section should be added.

Author Response

We want to thank the reviewers for the valuable and positive comments.

We would like to clarify that the modifications presented by us in M&M include the equipment and slight alterations on primes and HRM master Mix concentrations. These are mandatory adaptations to adequately test whether the typing outcome previously published could also be attained in our laboratory and routine conditions, as a reference center and biological Leishmania collection. Our aim was not to only and restrictedly confirm the findings published by Zampieri et al., 2016. On the contrary, we found the results well-grounded and quite promising. Therefore, we proposed to extend and complement the work initiated by that group to apply the approach in our routine, and for that such adaptations are unavoidable.  

A Conclusion is presented within the final part of the discussion. Since the IJMS does not consider a separate section mandatory, we would prefer to keep the submitted version.